# Clinical Progression and Outcome of Hospitalized Patients Infected with SARS-CoV-2 Omicron Variant in Shanghai, China

**DOI:** 10.3390/vaccines10091409

**Published:** 2022-08-28

**Authors:** Jiasheng Shao, Rong Fan, Jianrong Hu, Tiejun Zhang, Catherine Lee, Xuyuan Huang, Fei Wang, Haiying Liang, Ye Jin, Ying Jiang, Yanhua Gu, Gang Huang

**Affiliations:** 1Department of Infectious Diseases and Immunology, Jiading District Central Hospital Affiliated Shanghai University of Medicine & Health Sciences, Shanghai 201899, China; 2Genomics, Biotechnology Center, Center for Molecular and Cellular Bioengineering, Technische Universität, 01307 Dresden, Germany; 3Department of Respiratory Medicine, Jiading District Central Hospital Affiliated Shanghai University of Medicine & Health Sciences, Shanghai 201899, China; 4Department of Epidemiology, School of Public Health, Fudan University, Shanghai 200032, China; 5College of Public Health, University of Georgia, Athens, GA 30602, USA; 6Department of Urology, Renji Hospital, Shanghai Jiaotong University, Shanghai 200127, China; 7Department of Intensive Care Unit, Jiading District Central Hospital, Shanghai 201899, China; 8Department of Nursing, Jiading District Central Hospital, Shanghai 201899, China; 9Shanghai Key Laboratory of Molecular Imaging, Shanghai University of Medicine and Health Sciences, Shanghai 201318, China

**Keywords:** SARS-CoV-2, Omicron variant, vaccination, PCR conversion, Paxlovid

## Abstract

Background: Studies on the Omicron variant infection have generally been restricted to descriptions of its initial clinical and epidemiological characteristics. We investigated the timeline-related progression and clinical outcome in hospitalized individuals with the Omicron variant. Methods: We conducted a retrospective, single-centered study including 226 laboratory-confirmed cases with the Omicron variant between 6 April and 11 May 2022 in Shanghai, China. The final date of follow-up was 30 May 2022. Results: Among 226 enrolled patients, the median age was 52 years, and 118 (52.2%) were female. The duration from onset of symptoms to hospitalization was 3 days (interquartile range (IQR): 2–4 days) for symptomatic patients. Cough occurred in 168 patients (74.3%). The median interval to negative reverse-transcriptase PCR tests of nasopharynx swab was 10 days ((IQR): 8–13 days). No radiographic progressions were found in 196 patients on the 7th day after onset of symptoms. The median duration of fever in all participants was 5 days (IQR: 4–6 days). The median PCR conversion time of Paxlovid-treated patients was 8 days (IQR: 7–10 days) compared with that of a traditional Chinese herb medicine lianhuaqingwen (10 days, IQR: 8–13 days) (*p* = 0.00056). Booster vaccination can significantly decrease the severity of Omicron infection when compared with unvaccinated patients (*p* = 0.009). In multivariate logistic analysis, erythrocyte sedimentation rate (ESR) (OR = 1.05) was independently related to the severity of the infection. Conclusions: The majority of clinical symptoms of Omicron infection were not severe. Early and aggressive administration of Paxlovid can significantly reduce the PCR conversion time. Booster vaccination should also be highly recommended in the population over 14 years old.

## 1. Introduction

The novel Omicron variant was first reported to the World Health Organization (WHO) by the Gauteng province, South Africa, in mid-November last year and shortly after was classified as a variant of concern (VOC) [1]. The novel variant shares several mutations with the previous Alpha, Beta, and Gamma VOCs, which spread over more than 100 countries and immediately raised international concerns due to its higher transmissibility and infectivity [2,3].

Clinical characteristics of the Omicron variant have been partially described. Kim et al. reported 40 cases, and Lee et al. reported 83 cases of SARS-CoV-2 Omicron in South Korea, respectively, suggesting that patients in their studies had no clinical symptoms or mild symptoms and had recovered within several days [4,5]. Another report with 43 patients recruited from Center for Disease Control and Prevention (CDC) of the U.S. indicated that the most commonly reported symptoms were cough, fatigue, and rhinorrhea. Only one vaccinated individual was hospitalized for 2 days, and no deaths were reported in their study [6].

Since August 2021, China adopted a dynamic zero-COVID strategy to respond to SARS-CoV-2 variants with higher transmissibility and successfully stemmed hundreds of COVID-19 outbreaks that are associated with imported cases [7]. Even under the local administration’s aggressive containment, the variant spread very fast across Shanghai since mid-March 2022 and arrived at its peak on 13 April 2022, with about 25,000 infected individuals in one single day (Figure 1). As of May, 617,979 people in Shanghai had been infected [8]. However, the clinical course of the Omicron is still not fully described. 

Paxlovid (nirmatrelvir + ritonavir) raises new hopes of COVID-19 recovery in the age of the Omicron variant. It is a novel molecular entity recognized as a promising broad-spectrum agent, which can be used to treat infections with multiple human coronaviruses in vitro [9]. Lianhuaqingwen is a Chinese medicine, which has been routinely used in influenza therapy for decades and plays a role in the treatment of various influenza viruses. Shen et al. proved that it can significantly improve laboratory results of patients with COVID-19 and could be effectively applied alongside standard treatment of patients with moderate-type COVID-19 [10]. A meta-analysis showed that lianhuaqingwen combined with conventional treatment is more effective for patients with mild or moderate COVID-19 [11]. However, there is still a lack of clinical information on their efficacy for patients with the Omicron variant.

Herein, we collected data from 226 laboratory-confirmed cases in our hospital as 1 of 44 designated hospitals for treating SARS-CoV-2 Omicron in Shanghai, aiming to present the clinical progress and outcome of the Omicron variant infection. The demographic distribution of Omicron-variant-infected people across Shanghai is shown in Figure 2.

## 2. Methods

### 2.1. Study Designs and Participants

For this retrospective, single-center study, we recruited patients from 18 April 2022 to 11 May 2020 at Jiading District Central Hospital, Shanghai, China, an officially-designated hospital that treats patients with the Omicron variant. Symptomatic patients with the Omicron variant in this study were enrolled from mobile cabin hospitals, fever clinics, and nursing homes. There were four asymptomatic patients also enrolled because of unstable clinical symptoms such as dizziness and sudden elevation of blood pressure. Criteria for admission, diagnoses, therapy, and discharge were made based on the ninth trial version of COVID-19 pneumonia diagnosis and treatment plan (shortened as COVID-19 Treatment Plan) [12]. This study was approved by the Ethics Committee of Jiading District Central Hospital, and informed consent was obtained from all patients involved before data were collected.

### 2.2. Procedures

Information of demographic, epidemiological, clinical symptoms and signs, underlying co-existing disorders, laboratory and radiological findings, treatment, and outcomes were collected from medical records, and radiological images were scrutinized by a well-trained team of physicians. In clinical settings, the date of onset of disease was defined as the day when the symptom(s) was presented. Negative PCR conversion of the upper respiratory tract was defined as an interval between the first date of PCR positivity and the first date of two consecutively negative days by testing nasopharyngeal swab. Body temperature was measured by nurses using mercury thermometers at least four times a day. Fever was defined as an axillary temperature of 37.5 °C or higher [13]. The definition of defervescence was if the body temperature could be maintained normally without administration of any antipyretics. Based on the COVID-19 Treatment Plan, the disease severity was defined as asymptomatic: Omicron variant nucleic-acid-positive confirmed by reverse transcription polymerase chain reaction (RT-PCR) without clinical symptoms; mild: mild clinical manifestations such as fever, cough, expectoration, fatigue, etc., and without radiologic signs of pneumonia; moderate: clinical signs mentioned above, with radiologic signs of pneumonia; severe: clinical manifestations mentioned above, with radiologic signs of pneumonia plus anyone of the following: (1) shortness of breath and respiration rate (RR) ≥ 30 times/min, (2) the oxygen saturation (SaO_2_) is ≤93% in the state of resting and inhalation of air, (3) PaO_2_/FiO_2_ ≤300 mmHg (1 mmHg = 0.133 kpa) and PaO_2_/FiO_2_ shall be corrected according to the following formula in high altitude areas (more than 1000 m above sea level): PaO_2_/FiO_2_ × [760/atmospheric pressure (mmHg)], (4) progressive aggravation of clinical symptoms mentioned above and chest computed tomography (CT) showing significant progression of lesions >50% within 24~48 h; and critical: (1) respiratory failure occurs and mechanical ventilation is required, (2) shock, (3) and patients with other organ failure needing ICU monitoring and treatment [12]. We used vaccination doses to represent the level of immunization of the patients acquired before the Omicron variant infection.

All patients were admitted to isolation wards. Patients over 14 years were given supportive treatment after admission. We did not prescribe medicines to patients under 14 years due to their mild clinical symptoms and recommendations by COVID-19 Treatment Plan. Paxlovid (nirmatrelvir 300 mg and ritonavir 100 mg, twice per day) was used in 17 patients (13.0%) over 50 years old upon their admission for 5 consecutive days. Lianhuaqingwen (6 g, three times per day) was used in 114 patients (87%) above 14 years old upon their hospitalization, respectively. In the Paxlovid-treatment group, there were 10 patients with mild clinical symptoms and 7 patients with moderate clinical symptoms. Meanwhile, there were 95 mild cases and 19 moderate cases in the Lianhuaqingwen-treatment group. Currently, Paxlovid has not been regularly used in clinical practice in China, which is the reason why only a relatively small sample size was included in the analysis. Lianhuaqingwen, as a Chinese traditional medicine, was prescribed to patients according to the COVID-19 Treatment Plan [12]. The evaluation conditions for these two drugs are the period of Omicron viral PCR-negative conversion. Indications for antibacterial therapy of forty-two patients included fever and laboratory findings such as elevations of white blood cells (WBC), neutrophils, C-reactive protein (CRP), and procalcitonin. The selection of antibiotics (moxifloxacin, 18 (42.9%); ceftriaxone, 24 (57.1%)) was based on clinical experience and drug sensitivity test. The effect of two drugs on the evolution of the disease were evaluated according to defervescence and laboratory findings including normalization of WBC, neutrophils, CRP, and procalcitonin.

### 2.3. Laboratory Confirmation

Laboratory confirmation of the Omicron variant infection was accomplished by the Shanghai Center for Disease Prevention and Control (CDC). Subsequent test of nasopharyngeal swab specimen for the variant after hospitalization was performed by both Jiading District Central Hospital and Shanghai CDC based on the recommendation by the National Institute for Viral Disease Control and Prevention (China).

PCR procedure for identification of the Omicron variant was performed as described below: nasopharyngeal swab samples were collected for extracting viral RNA from patients suspected of having Omicron variant infection. After collection, the nasopharyngeal swabs were placed into a collection tube with 200 μL of virus preservation solution, and total RNA was extracted within 2 h using the respiratory sample RNA isolation kit (Shanghai ZJ Bio-Tech Co., Ltd., Shanghai, China). The reaction mixture contains 12 μL of reaction buffer, 4 μL of enzyme solution, 4 μL of Probe primers solution, and 5 μL of RNA template. Two target genes were amplified: Target 1—open reading frame 1 ab (ORF1ab): Forward primer: CCCTGTGGGTTTTACACTTAA, Reverse primer: ACGATTGTGCATCAGCTGA, Probe: 5′-FAM-CCGTCTGCGGTATGTGGAAAGGTTATGG-BHQ1-3′. Target 2—nucleocapsid protein (N): Forward primer: GGGGAACTTCTCCTGCTAGAAT, Reverse primer: CAGACATTTTGCTCTCAAGCTG, Probe: 5′-FAM-TTGCTGCTGCTTGACAGATT-TAMRA-3′. The cycling conditions for the amplification of ORF1ab and N genes: incubation at 50 °C for 10 min and 95 °C for 5 min, 45 cycles of denaturation at 95° for 10 s, annealing and extension at 55° for 40 s. A cycle threshold value (Ct-value) less than 35 was defined as a positive test result, and a Ct-value of 40 or more was defined as a negative test, while a Ct-value of 35 to 40 required confirmation by retesting [12].

### 2.4. Statistical Analysis

Means and interquartile range (IQR) for continuous variables were compared via independent group *t*-tests. Frequencies and proportions for categorical variables were compared with the chi-square test. The outcome of this study was measured as PCR conversion days. The outcomes were compared between different possible affecting variables by Kaplan–Meier curves. The association between baseline clinical characteristics, laboratory findings, and factors associated with the severity of infection were calculated with logistical regression. All statistical analyses were performed using SAS software version 9.4 (SAS institute, Cary, NC, USA). A *p*-value < 0.05 was considered statistically significant.

## 3. Results

### 3.1. Clinical and Laboratory Characteristics of Participants on Admission

As of 11 May 2022, the study population involved 226 hospitalized patients including 28 children with confirmed Omicron variant infection. The median age was 52 years old (IQR, 32–68 years) and 118 (52.2%) were female. Ninety-two (40.7%) patients were unvaccinated; 7 (3.1%), 56 (24.8%), and 71 (31.4%) patients received one dose (partially), two doses (fully), and three doses (booster) vaccination, respectively, with CoronaVac (SARS-CoV-2 Inactivated Vaccine (Vero Cell)) manufactured by Sinovac Biotech Co., Ltd. (Beijing, China) The median duration from onset of symptoms to hospital admission was 3 (2–3) days in patients with symptoms. In total, 105 patients (46.5%) had one or more chronic comorbidities. Hypertension was the most common coexisting disorder (76 (33.6%)), followed by type 2 diabetes (30 (13.3%)). Most common symptoms on admission are cough (168 (74.3%)), sputum production (111 (49.1%)), and fever (103 (45.6%)). Less common symptoms included sore throat, fatigue, dizziness and headache, rhinorrhea, loss of olfaction, myalgia/arthralgia, diarrhea, and shortness of breath (Table 1).

Table 2 shows the laboratory findings on admission. Lymphocytopenia was present in 35.6% of the patients, thrombocytopenia in 15.5%, and leukopenia in 19.1%. Erythrocyte sedimentation rate (ESR) was increased in 22.3% of the patients, while CRP was elevated in 24% of the patients. Elevated levels of alanine aminotransferase and aspartate aminotransferase were less common. D-dimers were increased in 30% and ferritin in 26.8% of the participants, respectively.

Shown are the official data of all documented, laboratory-confirmed cases of the Omicron variant throughout Shanghai, according to the Shanghai Municipal Health Commission as of 11 May 2022. The numerator indicates the number of patients who were included in the cohort study, and the denominator indicates the number of cases with clinical symptoms for each district, as reported by the Shanghai Municipal Health Commission.

### 3.2. Clinical Outcomes of the Study Population

The median interval of PCR conversion for Paxlovid-treated patients was 8 days (IQR: 7–10 days) compared to that in Lianhuaqingwen-treated patients (10 days, IQR: 8–13 days) (*p* = 0.00056) (Figure 3). To the time of submission, a total of 224 (99.1%) patients were discharged after 14 (12–17) days hospitalization, and two patients were still in the intensive care unit (ICU). No patients perished.

### 3.3. Duration of Fever in the Study Population

As fever was one of the cardinal signs among patients with COVID-19 [14], we investigated the defervescence period in this population. Fever occurred in 103 (45.6%) patients during their clinical course. Body temperature gradually returned to normal with or without supportive therapy after hospitalization. The median duration of fever in all participants was 5 days (IQR: 4–6 days) after onset of symptoms. The median duration of fever in moderate patients was 6.5 days (IQR: 6.0–8.3 days). Two severe cases including a 90-year-old male patient who developed from moderate to severe had significantly longer duration of fever (14 days, IQR: 12–16 days) (Figure 4). Most other symptoms, including cough, expectoration and sore throat, fatigue, dizziness, and headache, also relinquished before fever abatement.

### 3.4. Imaging Changes in Disease Progression

Chest computed tomography (CT) or X-ray were performed for 198 patients excluding 28 children upon admission. In all, 11 patients showed unilateral lesions, 31 patients were manifested with bilateral lesions, while the remaining 156 adult patients were normal in imaging. Radiological examinations were repeated in 198 patients; no progressions were shown in 196 cases (99.0%) after 7 days of symptoms onset. Only two senile male patients deteriorated in imaging and their clinical conditions. The representative dynamic imaging changes of three patients with different severities are shown in Figure 5.

### 3.5. Viral Eradication in the Upper Respiratory Tract

All patients had daily PCR testing for the Omicron variant in upper respiratory tract samples during this observation period. The median interval of PCR-negative conversion was 10 days (IQR: 8–13 days) in all patients. In four asymptomatic patients, it took them 9 days to be viral RNA-negative after admission (IQR: 8–10.5 days). The median time of PCR conversion was significantly longer in patients with moderate illness than those with mild (13 days, IQR: 10–15 days, vs. 10 days, IQR:8–12.5 days (*p* = 0.031) (Figure 6). The PCR conversion time in patients under 14 was significantly shorter than patients above 14 (8 days, IQR: 7–9 days vs. 11 days, IQR: 9–14 days, *p* <0.0001) (Figure 7).

### 3.6. Factors Related to the Severity of the Infection

To figure out factors that were associated with the severity of the disease, we compared clinical laboratory characters and vaccination status of asymptomatic (*n* = 4), mild (*n* = 180), and moderate (*n* = 41) patients.

In univariate analysis, older age, comorbidity, lymphopenia, high levels of CRP, ESR, lactate, estimated glomerular filtration rate (e-GFR), low levels of albumin, and fewer vaccination doses are all associated with the severity of Omicron infection. In multivariate logistic analysis, ESR (OR = 1.05) was independently related to the severity of the infection (Table 3).

## 4. Discussion

On 24 November 2021, the World Health Organization (WHO) announced a new variant of SARS-CoV-2 Omicron in South Africa, and 17 days later, the first case infected with Omicron in China was identified [15]. Recent investigations on the Omicron variant have described the epidemiologic characteristics, initial clinical, laboratory, and radiological findings [4,6,16]. To our knowledge, we are the first to describe the temporal clinical progression of Omicron variant infection to date. Our investigation has several distinguished features from current research. As one of the megacities and most important economic centers in China, the policy of “Dynamic zero” was implemented strictly in Shanghai. The whole city underwent completely lockdown during the pandemic outbreak since the end of March 2022 [17]. Under the policy of “all those in need have been tested, and if positive, have been quarantined, hospitalized, or treated” in China, 25 million residents were tested daily for Omicron strain infection by RT-PCR. All positive patients without symptoms were transferred to mobile cabin hospitals. Those symptomatic patients who had potential to worsen were directly sent to the fever clinic or COVID-19-designated hospital. All participants in our study came from 16 districts across Shanghai and were sent to our hospital from mobile cabin hospitals, fever clinics, and nursing homes, which might represent the current real-world population.

In this study, most infection by Omicron were mild or moderate and did not aggravate into severe or critical conditions. A recent investigation showed Omicron can cause lower severity disease than the Delta variant, which may be attributed to the fact that Omicron replicates less efficiently in the lung and more efficiently in the respiratory tract [18]. However, it has high transmissibility, which led to nearly 0.62 million Shanghai residents being infected in 2 months. To prevent and control the Omicron infection and its related diseases, exploring the clinical progression and factors associated with the severity of the infection is needed.

In accordance with a previous study from the U.S. CDC [6], we found that as high as 74.3% of the patients with the variant were suffering from cough, which indicated that the main attacking site of the virus is located on the upper airways [19]. Coughing can generate high speeds of aerosol containing viral particles, which can be expelled easily from the nose and mouth. This could explain why the variant can spread so fast in such a short period of time [20]. There is an investigation confirming that covering the mouth and nose for SARS-CoV-2-infected patients can speed PCR conversion since they contracted a smaller virus load [21]. Therefore, wearing a face mask is strongly recommended for stopping the higher transmissibility of the Omicron variant.

The infectiousness and transmissibility of SARS-CoV-2 has been associated with viral shedding [22,23]. To date, there are few studies on the predictors of the Omicron variant’s shedding duration. An interpretation for Omicron variant eradication in the upper respiratory tract is very important for the implementation of preventive tactics such as determination of the isolation period. Data from Japan suggest that the amount of viral RNA reached its highest three to six days after diagnosis or symptom emergence [24]. The median time after onset of symptoms to viral clearance in this study was 10 days, which was close to that of the COVID-19 original strain [25]. In addition, no radiographic progressions were identified in 98% of patients after hospitalization, which could be attributed to its lower virulence and having less chance to cause damage in the lungs [26,27].

It was reported that administration of Paxlovid can reduce hospital admissions and deaths among individuals with high risk, especially for elderly. In our study, Paxlovid was also proven to significantly shorten the period of viral PCR conversion in patients whose conditions were mild and moderate. As a traditional Chinese medicine, lianhuaqingwen can significantly suppress SARS-CoV-2 replication affecting the morphology of viral granule and exerting antiviral activity in vitro [28,29]. A meta-analysis showed that it may improve the clinical symptoms of patients infected with SARS-CoV-2, such as fever, fatigue, and myalgia [11]. In this retrospective study, it did not shorten the interval of PCR conversion in mild cases in patients aged above 14 years compared with the Paxlovid-treated group. However, it probably can alleviate the clinical symptoms of Omicron variant infection including cough, expectoration, and rhinorrhea.

Although cough is the most common symptom in Omicron infection, fever is still an important indicator for uncontrolled viral replication in the body. The median duration of fever in all febrile patients was 5 days, which was significantly shorter than that in original strain of SARS-CoV-2 (10 days) and SARS (11.4 days), indicating its decreased virulence [25].

COVID-19-inactivated vaccine was the main type of vaccine used in China’s national inoculation program. Currently, 95.1% and 42.5% of Shanghai permanent residents received full and boosted vaccination, respectively. Only 39.3% of people aged above 60 received a booster vaccination [8]. Our study manifested that there was a significant difference regarding severity between patients who received booster vaccinations and those who were unvaccinated, partially vaccinated, or fully vaccinated. This result is consistent with research from Hongkong which confirmed that both boosted vaccination of inactivated and mRNA vaccines can provide sufficient protective capability against severe/fatal cases for all Omicron-infected individuals [30]. This phenomenon might be attributable to neutralizing antibodies elicited by vaccination. The discovery was in line with another finding stating unvaccinated patients infected with the preceding variants of SARS-CoV-2 showed occasional neutralization to the Omicron variant as well [31]. As the virus continues to evolve, we cannot predict the subsequent mutating direction of the virus and whether its causes will be more or less severe. The whole society must constantly adapt by increasing their immunity through vaccination [32].

A recent investigation by Butt A. et al. showed Omicron variant infection in children who did not have prior infection or vaccination is associated with less severe disease than the Delta variant infection [33]. Our findings also indicated that children with the Omicron variant have mild symptoms and need much fewer medical interventions. As already known, young children are especially vulnerable to upper airway infection due to their relatively narrow and collapsible nasal passages and that babies breathe only through their noses [34], which means the virus mainly lingers in the upper respiratory tract. However, at the same time, the virus can be relatively driven out by breathing or coughing. Nevertheless, the actual mechanism behind this phenomenon needs to be elucidated in future studies.

This study has several limitations. First, we did not test dynamic changes of IgM and IgG in Omicron-infected patients, so we cannot determine what protective degree the vaccinated patients have. Second, we only tested viral PCR conversion in nasopharyngeal and not in sera, stool, or urine, which can also affect viral shedding time (VST) [35,36]. However, a previous study showed that the VST of SARS-CoV-2 in feces was longer than that in respiratory tract samples, raising the concern that the virus could infect individuals by fecal–oral transmission [37]. Third, two patients were still hospitalized at the time of manuscript submission; therefore, clinical outcomes of the patients cannot be obtained, and continued follow-up is still needed.

In conclusion, the majority of the Omicron variant cases in this study are not severe. The disease progression suggests that population-based booster vaccination and early control of viral replication by application of Paxlovid are essential to improve the prognosis of Omicron variant infection, especially in older populations and those who are suffering from coexisting illnesses.

## Figures and Tables

**Figure 1 vaccines-10-01409-f001:**
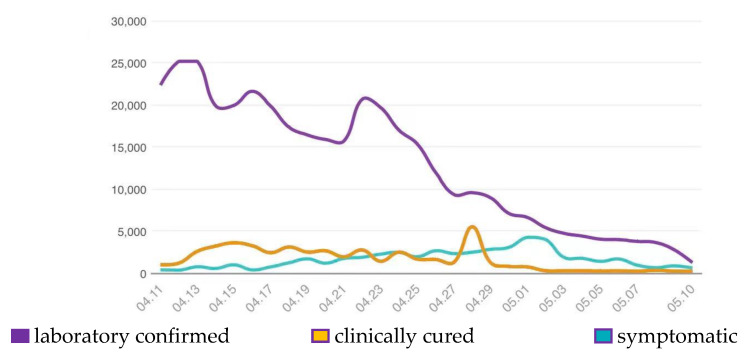
Legend of laboratory-confirmed, clinically cured, and symptomatic cases infected with the COVID-19 Omicron variant from 11 April to 10 May 2022.

**Figure 2 vaccines-10-01409-f002:**
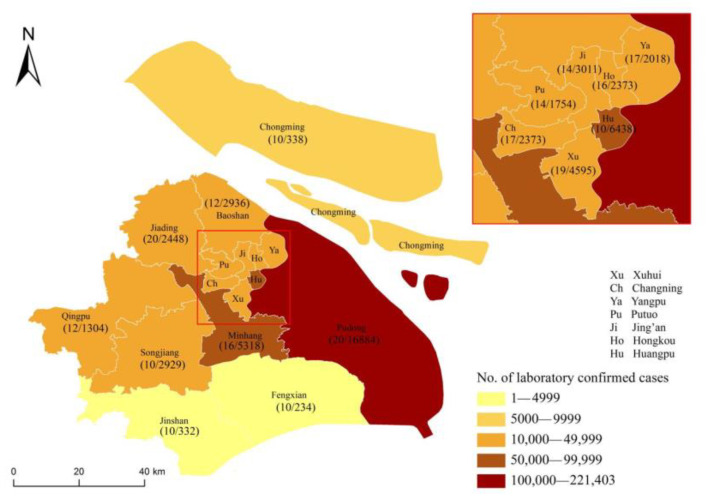
Distribution of patients with the Omicron variant across Shanghai from 1 March to 11 May 2022.

**Figure 3 vaccines-10-01409-f003:**
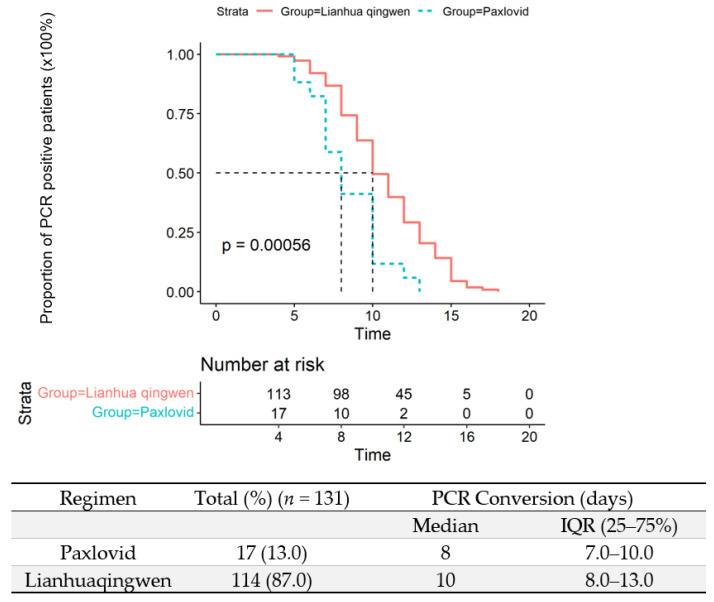
The comparison of PCR conversion time between Paxlovid- and Lianhuaqingwen-treated patients.

**Figure 4 vaccines-10-01409-f004:**
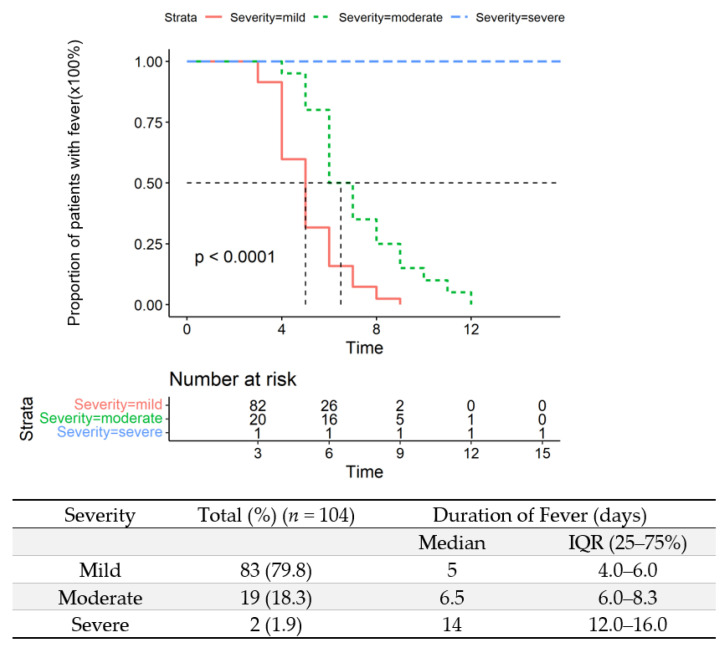
Duration of fever in mild, moderate, and severe cases.

**Figure 5 vaccines-10-01409-f005:**
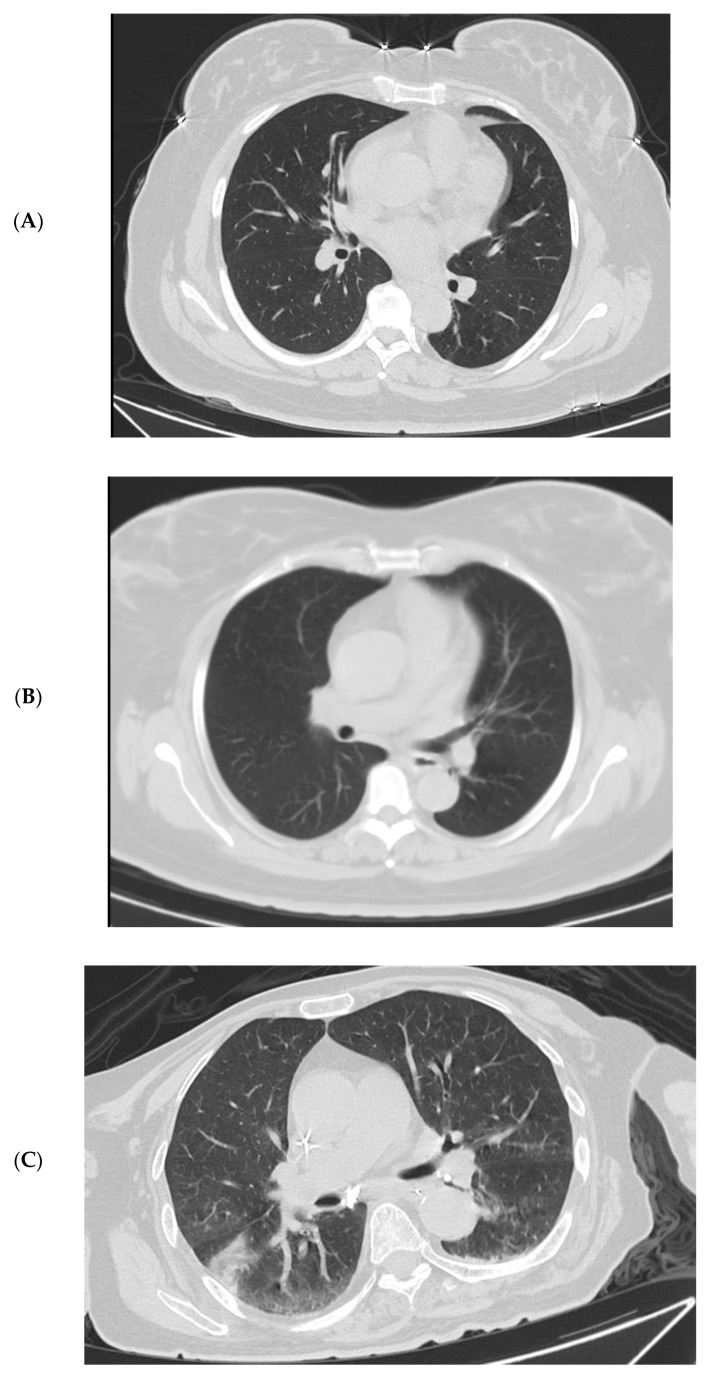
Representative chest radiographic manifestations in three patients with the Omicron variant. **Case 1:** Chest CT scans on admission of a 46-year-old female (mild) on day 2 after onset of symptoms (**A**). No new lesions were found on day 7 (**B**). **Case 2:** Chest CT scans on admission of a 77-year-old female (moderate) on day 2 after onset of symptoms (**C**). CT scan showed worsening on day 9 manifested by bilateral multiple patchy shadowing (**D**); chest CT scan on day 14 showed improved status (**E**). **Case 3:** Chest CT scans on admission of a 90-year-old male (severe) on day 5 after onset of symptoms bilateral lower lobes patchy shadowing (**F**). Bedside Chest X-ray on day 13 showed worsening status with bilateral diffuse patchy shadowing and consolidation (**G**).

**Figure 6 vaccines-10-01409-f006:**
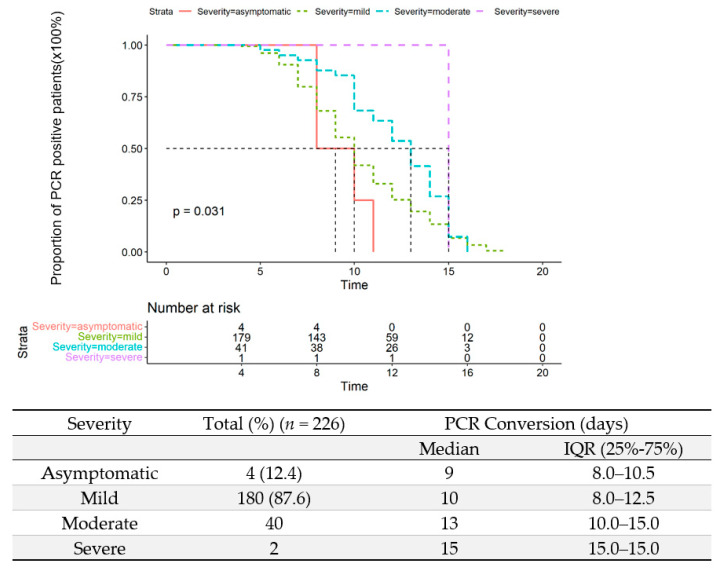
Time to a negative conversion of Omicron infection by PCR of upper respiratory tract samples among asymptomatic, mild, moderate, and severe cases.

**Figure 7 vaccines-10-01409-f007:**
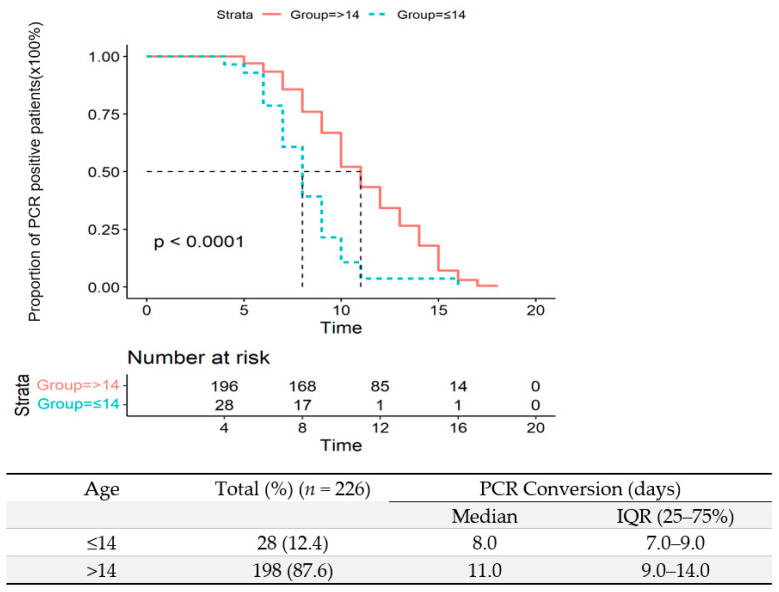
Time to a negative conversion of Omicron infection by PCR in patients under 14 and above 14.

**Table 1 vaccines-10-01409-t001:** Clinical characteristics of patients infected with the Omicron variant on admission (*n* = 226).

Characteristics	All Patients (*n* = 226)	Disease Severity				*p*-Value
		Asymptomatic (*n* = 4)	Mild (*n* = 180)	Moderate (*n* = 41)	Severe (*n* = 1)	
Age						
Median (IQR), y	52.0 (32.0–68.0)	51.5 (43.5–58.5)	48.0 (28.5–62.5)	71.0 (57.0–78.0)	78.0 (78.0–78.0)	<0.001
Distribution—No./total No. (%)						<0.001
0–14 y	28 (12.4)	0 (0.0)	28 (15.6)	0 (0.0)	0	
15–49 y	73 (32.4)	1 (25.0)	66 (36.9)	6 (14.6)	0	
50–64 y	52 (23.1)	2 (50.0)	42 (23.5)	8 (19.5)	0	
≥65 y	72 (32.0)	1 (25.0)	43 (24.0)	27 (65.9)	1 (100)	
Female sex—No./total No. (%)	118 (52.2)	2 (50.0)	99 (55.0)	17 (41.5)	0	0.251
Smoking history—No./total No. (%)						0.245
Smoker	27 (11.9)	0 (0.0)	21 (11.7)	6 (14.3)		
Non-smoker	199 (88.1)	4 (100.0)	159 (88.3)	36 (85.7)		
Median days from onset of symptoms to admission	3.0 (2.0–3.0)	/	3.0 (2.0–3.0)	3.0 (2.0–4.0)	2.0 (2.0–2.0)	0.608
Median vaccination doses (IQR)	2.0 (0.0–3.0)	3.0 (2.0–3.0)	2.0 (0.0–3.0)	0.0 (0.0–2.0)	0.0 (0.0–0.0)	0.009
Fever—No./total No. (%)	103 (45.6)	0 (0.0)	83 (46.1)	20 (48.8)	0 (0.0)	0.226
Median temperature (IQR)—°C	38.4 (38.0–39.0)	/	38.4 (38.0–39.0)	38.4 (37.9–38.8)		0.444
Duration of fever	5.0 (4.0–6.0)	/	5.0 (4.0–6.0)	6.5 (6.0–8.3)	14.0 (12.0–16.0)	<0.001
Distribution of Temperature—No./total No. (%)						0.280
<37.5 °C	123 (54.4)	4 (100.0)	97 (53.9)	21 (51.2)	1 (100.0)	
37.5–38.0 °C	35 (15.5)	0 (0.0)	28 (15.6)	7 (17.1)	0 (0.0)	
38.1–39.0 °C	51 (22.6)	0 (0.0)	38 (21.1)	13 (31.7)	0 (0.0)	
>39.0 °C	17 (7.52)	0 (0.0)	17 (9.4)	0 (0.0)	0 (0.0)	
Symptoms—No. (%)						
Cough	168 (74.3)	0 (0.0)	133 (73.9)	34 (82.9)	1 (100.0)	0.005
Sputum production	111 (49.1)	0 (0.0)	85 (47.2)	25 (61.0)	1 (100.0)	0.031
Loss of olfaction	16 (7.1)	0 (0.0)	14 (7.8)	2 (4.9)	0 (0.0)	0.822
Fatigue	33 (14.6)	0 (0.0)	27 (15.0)	6 (14.6)	0 (0.0)	1.000
Dizziness and headache	28 (12.4)	0 (0.0)	22 (12.2)	6 (14.6)	0 (0.0)	0.801
Shortness of breath	4 (1.8)	0 (0.0)	3 (1.67)	1 (2.44)	0 (0.0)	0.600
Rhinorrhea	18 (8.0)	0 (0.0)	16 (8.89)	2 (4.88)	0 (0.0)	0.696
Sore throat	44 (19.5)	0 (0.0)	40 (22.2)	4 (9.76)	0 (0.0)	0.212
Diarrhea	6 (2.7)	0 (0.0)	3 (1.67)	3 (7.32)	0 (0.0)	0.196
Myalgia/arthralgia	11 (4.9)	0 (0.0)	8 (4.44)	3 (7.32)	0 (0.0)	0.559
Comorbidity—No. (%)						
Respiratory system diseases	6 (2.7)	0 (0.0)	4 (2.2)	2 (4.9)	0 (0.0)	0.396
Type 2 diabetes	30 (13.3)	0 (0.0)	18 (10.0)	12 (29.3)	0 (0.0)	0.012
Hypertension	76 (33.6)	1 (25.0)	50 (27.8)	24 (58.5)	1 (100.0)	<0.001
Non-hypertension cardiovascular disease	13 (5.8)	0 (0.0)	10 (5.6)	3 (7.3)	0 (0.0)	0.787
Cerebrovascular disease	22 (9.7)	0 (0.0)	9 (5.00)	13 (31.7)	0 (0.0)	<0.001
Hepatitis B virus infection	1 (0.4)	0 (0.0)	1 (0.6)	0 (0.0)	0 (0.0)	1.000
Cancer	14 (6.2)	0 (0.0)	6 (3.3)	7 (17.1)	1 (100.0)	0.001
Chronic renal disease	2 (0.9)	0 (0.0)	2 (1.1)	0 (0.0)	0 (0.0)	1.000
Endocrine disorder	2 (0.9)	0 (0.0)	2 (1.1)	0 (0.0)	0 (0.0)	1.000
Asthma	2 (0.9)	0 (0.0)	1 (0.6)	1 (2.4)	0 (0.0)	1.000
Parkinson’s disease	2 (0.9)	0 (0.0)	1 (0.6)	1 (2.4)	0 (0.0)	0.366
Immunodeficiency	1 (0.4)	0 (0.0)	0 (0.0)	0 (0.0)	1 (100.0)	0.204

**Table 2 vaccines-10-01409-t002:** Laboratory findings of enrolled patients on admission (*n* = 226).

Laboratory Findings	Reference Range	Median (IQR), on Hospital Admission (*n* = 226)	Median (IQR)				*p*-Value
			Disease Severity				
			Asymptomatic (*n* = 4)	Mild (*n* = 180)	Moderate (*n* = 41)	Severe (*n* = 1)	
WBC (×10^9^/L)	3.5–9.5	4.6 (3.7–6.0)	5.4 (4.6–6.1)	4.5 (3.7–5.9)	5.20 (3.7–6.2)	3.30 (3.3–3.3)	0.432
Hb (g/L)	130–175	130.0 (121.0–141.0)	138.0 (131.0–143.0)	130.0 (122.0–141.0)	124.0 (116.0–135.0)	103.0 (103.0–103.0)	0.052
PLT (×10^9^/L)	125–350	188.0 (139.0–227.0)	180.0 (154.0–192.0)	188.0 (140.0–232.0)	174.0 (138.0–224.0)	196.0 (196.0–196.0)	0.618
Lymphocyte (×10^9^/L)	1.1–3.2	1.3 (0.9–1.7)	1.1 (0.9–1.6)	1.3 (1.0–1.7)	1.1 (0.9–1.4)	0.6 (0.6–0.6)	0.034
Ferritin (ng/mL)	4.63–204	111.0 (47.0–218.0)	122.0 (81.8–203.0)	99.3 (40.2–187.0)	238.0 (142.0–336.0)	296.0 (296.0–296.0)	<0.001
ESR (mm/h)	<15	11.0 (8.0–15.0)	8.0 (7.3–10.8)	10.0 (8.0–12.0)	20.0 (14.0–29.0)	24.0 (24.0–24.0)	<0.001
CRP (mg/L)	0–8	2.9 (0.4–7.0)	3.4 (0.7–7.7)	2.0 (0.4–5.5)	9.7 (3.8–24.6)	71.7 (71.7–71.7)	<0.001
PCT (ng/mL)	≤0.25	0.0 (0.0–0.1)	0.1 (0.0–0.1)	0.0 (0.0–0.1)	0.1 (0.0–0.2)	1.3 (1.3–1.3)	<0.001
IL-6 (pg/mL)	0–10	4.7 (3.6–8.5)	6.6 (3.9–10.1)	4.2 (3.3–6.4)	13.8 (5.9–24.5)	23.6 (23.6–23.6)	<0.001
D-dimer (mg/L)	0–0.5	0.3 (0.2–0.6)	0.2 (0.2–0.3)	0.3 (0.2–0.5)	0.6 (0.3–1.2)	1.4 (1.4–1.4)	<0.001
TG (mmol/L)	0–1.8	1.2 (0.9–1.5)	1.0 (1.0–1.2)	1.2 (0.9–1.5)	1.2 (0.8–1.8)	1.1 (1.1–1.1)	0.841
TC (mmol/L)	2.84–6.2	3.8 (3.2–4.4)	3.6 (3.5–3.9)	3.8 (3.1–4.3)	4.1 (3.7–4.6)	3.1 (3.1–3.1)	0.370
HDL (mmol/L)	1.04–1.68	1.1 (0.9–1.4)	1.4 (1.1–1.5)	1.2 (1.0–1.5)	1.0 (0.9–1.1)	1.1 (1.1–1.1)	0.004
LDL (mmol/L)	≤3.3	2.3 (1.9–2.7)	2.1 (1.8–2.4)	2.3 (1.9–2.8)	2.3 (1.8–2.7)	1.4 (1.4–1.4)	0.500
Total bilirubin (umol/L)	3.4–20.5	8.3 (6.3–12.8)	11.8 (9.8–17.8)	7.9 (6.3–11.7)	9.3 (7.3–15.4)	17.6 (17.6–17.6)	0.031
Pro-BNP (pg/mL)	≤589	36.9 (27.5–93.0)	35.6 (27.7–40.5)	33.5 (25.5–57.0)	89.8 (47.9–222.0)	255.0 (255.0–255.0)	<0.001
ALT (IU/L)	<50	15.0 (12.0–23.0)	11.5 (8.0–16.2)	16.0 (12.0–23.0)	15.0 (12.0–25.0)	11.0 (11.0–11.0)	0.404
AST (IU/L)	17–59	23.0 (18.0–30.0)	17.0 (14.8–21.0)	22.0 (18.0–30.0)	24.0 (18.0–29.0)	45.0 (45.0–45.0)	0.222
Albumin (g/L)	35–53	39.0 (37.0–41.0)	41.0 (40.2–41.2)	40.0 (38.0–41.0)	36.0 (33.0–39.0)	28.0 (28.0–28.0)	<0.001
LDH (IU/L)	50–240	157.0 (132.0–201.0)	156.0 (131.0–183.0)	154.0 (129.0–194.0)	198.0 (156.0–255.0)	675.0 (675.0–675.0)	<0.001
Lactate (mmol/L)	0.7–2.1	1.1 (0.9–1.3)	1.1 (0.9–1.3)	1.0 (0.8–1.2)	1.4 (1.2–1.7)	2.0 (2.0–2.0)	<0.001
e-GFR (mL/min/1.73 m^2^)	46–92	99.6 (87.5–117.0)	98.5 (87.3–109.0)	103.0 (90.9–121.0)	85.4 (76.1–97.2)	92.2 (92.2–92.2)	<0.001
Potassium (mmol/L)	3.5–5.1	4.0 (3.7–4.3)	3.8 (3.8–3.9)	4.0 (3.7–4.3)	3.9 (3.5–4.4)	3.7 (3.7–3.7)	0.552
Sodium (mmol/L)	135–147	139.0 (137.0–141.0)	139.0 (139.0–140.0)	139.0 (137.0–141.0)	139.0 (137.0–141.0)	139.0 (139.0–139.0)	0.969
Chloride (mmol/L)	98–107	104.0 (101.0–106.0)	105.0 (103.0–106.0)	104.0 (101.0–107.0)	105.0 (102.0–106.0)	103.0 (103.0–103.0)	0.905

Abbreviation: WBC, white blood cell; Hb, hemoglobin; PLT, platelet; ESR, erythrocyte sedimentation rate; CRP, C-reactive protein; PCT, procalcitonin; IL, interleukin; TG, triacylglycerol; TC, total cholesterol; HDL, high-density lipoprotein; LDL, low-density lipoprotein; Scr, serum creatinine; BNP, B-type natriuretic peptide; ALT, alanine aminotransferase; AST, aspartate aminotransferase; LDH, lactic acid dehydrogenase; e-GFR, estimated glomerular filtration rate.

**Table 3 vaccines-10-01409-t003:** Factors related to the severity of the Omicron infection.

	Univariate Logistic Regression			Multivariate Logistic Regression		
	OR	95%CI	*p*-Value	OR	95%CI	*p*-Value
Age	1.06	1.03–1.08	<0.001	1.03	0.99–1.08	0.138
Female	0.58	0.29–1.15	0.120	0.51	0.20–1.24	0.143
Fever	1.11	0.56–2.19	0.757	1.97	0.75–5.35	0.172
Comorbidity	5.46	2.46–12.12	<0.001	1.11	0.33–3.69	0.867
WBC	1.09	0.94–1.26	0.266	0.95	0.74–1.20	0.674
Lymphocytes	0.48	0.25–0.91	0.025	0.70	0.27–1.65	0.459
CRP	1.03	1.01–1.05	0.004	0.99	0.96–1.02	0.387
ESR	1.08	1.04–1.13	<0.001	1.05	1.02–1.10	0.007
Albumin	0.81	0.73–0.89	<0.001	0.95	0.84–1.07	0.363
ALT	1.00	0.99–1.01	0.861	1.00	0.97–1.04	0.908
AST	1.00	0.99–1.00	0.920	1.00	0.97–1.02	0.703
Lactate	6.51	2.72–15.58	<0.001	2.34	0.67–8.25	0.180
e-GFR	0.97	0.95–0.98	<0.001	0.99	0.97–1.02	0.606
Dose of vaccinations						
0	1.00			1.00		
1	1.46	0.25–8.49	0.676	2.43	0.17–28.85	0.492
2	0.63	0.28–1.46	0.282	0.71	0.21–2.25	0.571
3	0.28	0.11–0.73	0.009	0.52	0.15–1.73	0.296

## Data Availability

Raw data used in this study, including de-identified patient metadata and test results, are available upon request. Reasonable requests to access the datasets should be directed to the corresponding author.

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
