# Peer review of "Clinical Progression and Outcome of Hospitalized Patients Infected with SARS-CoV-2 Omicron Variant in Shanghai, China"

_vaccines, 2022, doi:10.3390/vaccines10091409_

Round 1
Reviewer 1 Report
The authors describe a problem that is quite popular today, concerning infection with the Omicron variant of the SARS-CoV-2 virus. The new data on the clinical symptoms of covid-19 is very valuable and could be of use worldwide. Hence, the precision in giving various details and the use of names understandable to an international group of readers are crucial. In order to fulfill these conditions, the manuscript requires some corrections.
1. Please clearly sign figures !
2. What does the sentence mean: "Informed consent was waived for patients who were unable to obtain an informed consent."? If the research could also include those who were not able to consent to it, what is the value of the approval of the Ethics Committee? How many such patients were there?
3. Lines 93-94: According to what definition is this value taken as a fever?
4. I understand that only hospitalized patients were included in the study. If so, it would be good to clearly indicate it, e.g. in the title, because it informs the reader that the study concerns only a certain segment of the population infected with the Omicron variant.
5.Line 123: which type of diabetes? Please complete this information. "diabetes mellitus" is the old underlining, now the term "diabetes" is used and the type is given (e.g. type 2 diabetes-TD2).
6. In my opinion, there is no clear information that the study also included children (e.g. in lines 114-116) and no explanations why such a division of age groups was adopted (what were the criteria?), especially in groups 0-14 and 15-49 yr.
7. The abstract, the results and the discussion mention 2 medicines that were used in patients. Please organize and develop this topic. First of all: describe in the introduction both drugs, especially LHQW, because it is not a preparation commonly used in the world. Second, please refer to the results and correctly interpret the fact that Paxlovid was only used in 17 patients and LHQW was used in the majority of patients. In such a situation, can general conclusions be drawn about the effectiveness of both drugs?
Reviewer 2 Report
The study of the variants responsible for covid 19 remains important in order to better understand the morbidity and mortality related to it.
In their study, Shao et al investigated the evolution of symptoms and virus presence in a population heterogeneous in terms of age, immunization level and co-morbidity.
The methodology is very confusing and had considerable limits.
-The definition of disease severity is not clear in the methodology, although this is the criterion on which the authors based their definition of four groups in the results section. The only time the authors define these criteria is in the legend of Table 1. This is not the place to define the disease severity criteria.
Also, did the authors take into account the possible evolution of the severity of the disease over time? In fact, in Table 1, the authors reported only one severe case, whereas on page 6, lines 164-166, they indicated two severe cases.
- It is also described in the "results" section the efficacy data of two molecules used to control the disease. However, in the methodology section, there is no mention of the procedure for administering these treatments and the conditions of their evaluation remain unclear. For example, it is not known which category of the population benefited from this treatment as the whole study population is not concerned.
Furthermore, the results obtained with these molecules should be interpreted with caution, taking into account the severity of the disease, vaccination status, age and the sample size of the two groups receiving the treatment. Moreover, the relevance of this part with respect to the objective of the manuscript, which is to study the clinical evolution of patients infected with the omicron variant, is questionable. In addition, some individuals in the cohort were given antibiotic therapy. No indication is provided on the patients who received these antibiotics and their effect on the evolution of the disease.
The presentation of the results, especially the survival graphs, is problematic. Why does the proportion of survival decrease with time when in fact we see that the risk (the number of individuals at risk) decreases? Shouldn't we have the opposite on the survival graphs?
The authors made a point about the level of immunization and in the abstract indicated the impact that this could have on the degree of severity of the disease, however, neither in the methodology nor in the results, there is no data on this. They only indicated the distribution of their cohort according to vaccination status (page 3 lines 117-119) and in table 1 where they indicate the median time between the different vaccine doses.
Minor comments
page 3 lines 114-115 and figure 2 on page 5 do not belong in this section on study results
Page 5-6 (lines 145-154) should be included in the methodology
page 3, lines 97-102, it would be good to indicate the pcr procedure for identifying the omicron variant
Reviewer 3 Report
The article submitted for review shows some weaknesses.
On the one hand, the number of patients analyzed is very small, as there are studies in the literature with a much larger number of patients than those analyzed in the present work. The only difference shown by the authors in the work is the mode of health management of patient isolation during the pandemic in China, but in my opinion this is not relevant to the final conclusions, which can be applied to any country in the world with a different health management policy.
Likewise, the conclusions provided by the authors at the end of the paper are not very relevant (most of the Omicron infections are not very relevant, a vaccine booster is recommended for the population older than 14 years and Paxlovid can reduce the duration of the disease) and well known by the scientific community.
The article also has a large number of tables and figures that do not improve the reader's understanding of the article. I would remove tables 1-2-3 and figures 2-5.
I would not recommend the publication of the article in the journal as it does not provide new data on what is already known.
Reviewer 4 Report
The authors are describing several characteristics of patients infected with Omicron variant of SARS- Co-
V hospitalized in their center. Perhaps the term advancement both in title and the abstract is not the most appropriate.
Several English corrections are necessary because the meaning is not clear in several points. Furthermore discussion needs much more work to be done, for example comparison with MERS is not necessary. More details are included in the file attached.

Round 2
Reviewer 2 Report
The authors have considerably improved the quality of the manuscript.
However, as suggested earlier, some clinical data on patients who received both treatments are missing:
-why Paxlovid is only given to patients over 50 years of age. This information should be included in the manuscript (the authors just stated it in their response but not in the manuscript).
- similarly, they should specify why Lianhuaqingwen was only prescribed to patients over 14 years old.
- According to the authors, they tested these two therapies in order to provide more guidance to Chinese clinicians regarding possible treatments. However, they do not provide any information on the clinical status of the individuals who received the treatment. It would be useful to know if the clinical condition (mild, moderate, severe) affects the effectiveness of the treatments.
in the methodology, they should indicate that they made a comparison between two groups (>14 and <14 years old)
Reviewer 3 Report
In my opinion, the new version of the article submitted by the authors adequately responds to the request presented by the reviewers of the journal and could be accepted for publication in its current version.
Reviewer 4 Report
The authors have made a serious effort to correct their manuscript. Nevertheless several flaws and wrong English expressions have been used and need correction.
I am including the corrections in the text attached

Author Response
Dear reviewer,
Please see the attachment. Many thanks.
